# Handheld Ultrasound and Focused Cardiovascular Echography: Use and Information

**DOI:** 10.3390/medicina55080423

**Published:** 2019-07-31

**Authors:** Ketty Savino, Giuseppe Ambrosio

**Affiliations:** Cardiology University of Perugia, 06156 Perugia, Italy

**Keywords:** handheld ultrasound devices, diagnostic imaging, echocardiography, ultrasonography, point-of-care testing, focused echography, echoscopy

## Abstract

The availability of miniaturized ultrasound machines has changed our approach to many cardiovascular diseases. Handheld ultrasound imaging can be performed at the bedside, it is easy to use, and the information provided, although limited, is of unquestionable importance for a quick diagnosis that leads to early treatment. They have unique characteristics: Low cost, wide availability, safety, accuracy, and can be used in different clinical scenarios and by operators with different backgrounds. Image acquisition and interpretation is rapid and provides, in each situation, useful information for diagnosis, prognosis, and clinical and therapeutic management. This review focuses on the use of handheld ultrasound devices, describes differences with other equipment, their limitations, and the numerous advantages derived from their use.

## 1. Introduction

Cardiovascular ultrasound is an imaging investigation with unique characteristics: It is safe, low cost, widely available, repeatable, and accurate. In fact, it is the most used imaging technique in clinical practice. In cardiology, the use of various echocardiographic modalities allows a comprehensive study of cardiac structure, function, and hemodynamics. Mono-dimensional (M-mode), two-dimensional (2D), and three-dimensional (3D) echo define cardiac chamber dimensions and volumes, systolic function, and valvular morphology: Color and Doppler (pulsed and continuous) techniques allow accurate investigation of cardiac hemodynamics and diastolic function; tissue Doppler imaging (TDI) and two-dimensional strain (2DS) are able to detect clinical and subclinical systolic and diastolic dysfunction [1]. All derived echocardiographic information has a well-documented diagnostic and prognostic utility [2].

For many decades, echocardiography equipment was immobile, ultrasound scans were performed in specific echo-laboratories, and cardiologists were the only competent operators. With technological development, echocardiography equipment has become movable, portable, and miniaturized, and ultrasound use has become more extensive, at the bedside, in wider clinical scenarios, in critically ill patients, and in emergencies/urgencies, and its use has expanded to physicians with different backgrounds [3].

Nowadays, various types of equipment are available, with different sizes, different echo modalities, and different diagnostic capabilities. This review focuses on the use of handheld ultrasound devices (HUDs), and describes differences with other equipment, their limitations, and the numerous advantages that come from their use.

HUDs can be performed easily, rapidly, and allow basic information to be obtained for diagnosis and clinical management of various diseases; an HUD can be used by physicians from different disciplines, in many situations and clinical scenarios, especially in emergencies/urgencies and its use has transformed almost all aspects of our daily practice.

## 2. Differentiation of the Various Echocardiographic Equipment: From Standard, Comprehensive Transthoracic Echocardiography to Focused Cardiac Ultrasound

*Standard echocardiography* performed in echo-labs by cardiologists or sonographers/cardiologists provides information about cardiac size, structure, function, and hemodynamics. All these data are obtained using stationary systems that are equipped with various modality system and transducers: 2D, M-mode, Doppler (pulsed and continuous), color, and TDI, transesophageal approach (TEE), and, in many cases with advanced modalities, such as 2DS and three-dimensional echo (3D). During the exam, all modalities can be utilized (if necessary, also the advanced modalities), all standardized echo sections are performed, simultaneously with ECG-guided comprehensive measurements of cardiac structure, heart function, and hemodynamics. Thus, the final report is complete and accurate [1,2,4].

*Portable machines* are smaller, and allow a basic comprehensive exam of 2D, M-mode, pulse, and continuous Doppler and color. Generally, they do not have advanced modalities, the quality of images is good, and the exam is clinically complete. The final report describes all basic cardiac morphological and functional information.

*Handheld ultrasound* devices are the smallest machines, very simple to use, with a limited number of basic controls for adjusting the depth and gain, to freeze and store images (in JPEG) and little loops (in MPEG-4), and the available measurements are few and limited to a simple distance and area assessment. The devices only have the 2D modality with grey-scale images and color-Doppler, simultaneous ECG, M-mode, TDI, and advanced technologies are lacking. Images have lower spatial (640 × 480 pixels) and temporal resolution than other equipment. However, 2D and color Doppler are in real-time; the field-of-view (2D and color flow), maximum depth (25 cm), automatic frame rate (28 frames per second), and transducers (phased array—1.7 to 3.8 MHz) are similar; images have good technical quality; and a correct final diagnosis can be made in most cases (Figure 1) [5,6,7,8]. The exam is performed with a limited number of echo sections; often the evaluation is qualitative, with a bimodal (yes/no) or semi quantitative (normal/reduced) response. It represents an extension of the physical examination, is focused to recognize specific signs that lead to an answer to a clinical diagnostic suspicion in a specific clinical setting, and, because it is focused on getting a few findings, it is also named FoCUS [9,10] (Table 1).

## 3. Focused-Echo in Various Clinical Scenarios: When to Use It

FoCUS/HUDs can be used in many situations and different scenarios, with various acquisition protocols, in stable and unstable patients. It can be used in in- and out-patients and by operators with different specialties. Few measurements are possible, such as left and right atrium and ventricle dimensions, wall thickness, ascending aorta, and inferior vena cava (IVC), but HUDs can qualitatively provide useful information about atrial dilatation, left and right ventricle global systolic function, ventricular dilatation or hypertrophy, significant valvular stenosis and regurgitation, pericardial effusion, and tamponade [8,9,10]. In the literature, many studies have demonstrated that HUDs provide a more accurate diagnosis than physical examination for the majority of common cardiovascular diseases and FoCUS results correlate well with standard echocardiography [8,10,11,12,13,14,15,16,17].

In **stable patients,** FoCUS is appropriate for the screening of structural heart disease. It allows early diagnosis, defines prognostic stratification, and directs to the appropriate therapy. The exam can rapidly define/exclude qualitative ejection fraction, wall motion abnormalities, ventricular systolic dysfunction, left ventricular hypertrophy, cardiomyopathies with hypertrophic pattern, dilated cardiomyopathies, pulmonary congestion, significant valvulopathies, ascending aorta and aortic root dilatation, pericardial and pleural effusion, and IVC size and respiratory collapsibility [3,7,8,11,12,14,16] (Table 2).

Below are some examples of the effective use of HUD:

In patients with clinical suspicion of *congestive heart failure*, FoCUS helps to define atrial dilatation, wall motion abnormalities and thicknesses, cardiac dimension and function, significant mitral regurgitation, lung congestion (B-lines at lung ultrasound, also called “comets”), IVC collapse, and it can differentiate between systolic or diastolic heart failure (Figure 2) [18,19,20].

In *atrial fibrillation*, FoCUS defines left atrial dimensions (dilatation) and left ventricular function, which are strong predictors of sinus rhythm restoration and lead to proper treatment [21,22].

In patients with *valvular stenosis*, FoCUS can qualitatively show valvular morphology, such as thickened and calcified leaflets with reduced mobility and turbulent transvalvular flow by color Doppler. All these indirect signs lead to the suspicion of significant valvulopathy. Quantitative assessment is not possible, nor are pulmonary pressures, but in patients with signs/symptoms of heart failure, atrial fibrillation, or syncope, these data define the underlying pathophysiology [9,10,23].

In *valvular regurgitation*, color-HUDs can differentiate significant mild/trivial valvulopathy.

In patients with *pericarditis*, FoCUS can be used for basic follow-up of pericardial effusion (Figure 3) [9,20].

This approach is a screening tool that provides all information for diagnosis and immediate start for proper therapy, but it cannot be considered as a definitive investigation and standard echocardiography must follow later for a comprehensive heart disease evaluation [3,4,24].

In **unstable patients**, HUDs provide crucial information in critically ill patients or impending critical situations, is an essential technology for improving early diagnosis, help to rule-in/rule-out different pathological conditions, evaluates the pathophysiology of clinical status, and allows proper clinical management [15,25]. When an HUD is used at the point of care, the acronym FoCUS is changed to POCUS (point of care ultrasound).

For 20 years, many POCUS protocols have been proposed in the literature to standardize the procedure. They preferentially evaluate cardiac, pulmonary, and abdomen diseases, or are elective for searching for trauma lesions. Actually, because there are many clinical scenarios, data acquisition depends on the specific imaging target rather than specific protocol.

In *ill patients*, POCUS is extremely useful in terms of a differential diagnosis of clinical conditions that manifest with the same symptoms (Table 3).

In *acute respiratory failure*, lung ultrasound can differentiate pneumothorax from pleural effusion or cardiogenic and non-cardiogenic pulmonary edema [26].

In *shock patients*, POCUS is useful for diagnosis, management, and monitoring of treatment efficacy and clinical progression. Chamber size, left ventricular systolic function, IVC dimensions and collapsibility, pericardial effusion, and pulmonary congestion (lung B lines—comets) in lung ultrasounds identifies cardiogenic shock [27].

In *cardiac tamponade*, POCUS is useful to identify pericardial effusion size and distribution and directs to the best approach for pericardiocentesis [28].

In the *emergency department*, POCUS helps to differentiate the nature of chest pain. In addition to history, clinical examination, electrocardiogram, and biomarkers, echocardiography can visualize wall-motion abnormalities and left ventricular function, dilated right ventricle with free wall hypokinesia, ascending aorta dimension and morphology, significant aortic regurgitation, and pericardial effusion. In this way, it is a first step to a differential diagnosis among acute coronary syndromes, acute aortic syndromes, pulmonary embolism, congestive heart failure, and pericarditis [29].

In patients with *cardiac arrest* due to complex arrhythmias, HUDs can exclude/diagnose all cardiac arrhythmogenic diseases, such as hypertrophic, dilated, or arrhythmogenic right ventricular cardiomyopathies [30].

In *cardiopulmonary resuscitation* and during BLS (basic life support) and ALS (advanced life support) protocols, POCUS can help to diagnose potentially treatable causes of cardiac arrest, such as cardiac tamponade, massive pulmonary embolism, severe ventricular dysfunction, and hypovolemia [31,32].

## 4. Specialties That Use HUDs and Training of Operators

Since FoCUS and POCUS are simplified exams with restricted protocols, operators with different backgrounds can utilize them. Actually, cardiologists, internists, anesthesiologists, and specialists in emergency/urgency frequently utilize this tool. The approach is simplified; the findings to obtain are few and are guided by symptoms, so a short training period is required. Some studies have shown that brief training is sufficient to perform and interpret a FoCUS scan, the agreement between FoCUS and standard echocardiography is satisfactory, the trainee’s performance improves over a short period of time, and the inter-observer variability in FoCUS is low [33,34,35,36]. Nowadays, both European and American guidelines recommend a theoretical background on cardiovascular diseases and practical experience to obtain competence in image acquisition and in their interpretation [9,10]. The educational path is carried out with a tutor, who verifies correct image acquisition. At the beginning, data interpretation is discussed together and, subsequently, each operator individually acquires and interprets the data. Specific education and training in the use of HUDs is obtained by individually interpreting 25 to 50 exams [8,10,37].

## 5. Conclusions

The use of HUDs has completely changed the daily approach to cardiac diseases. Handheld devices have allowed fundamental information about morphology and cardiac function to be obtained in stable and, especially, unstable patients. They extend the physical examination with a short and simplified ultrasound examination aimed at achieving rapid diagnosis, early treatment, and basic monitoring of some cardiac diseases. Its use serves to guide diagnosis in specific clinical situations, mainly defined by patient symptoms; it is a limited and non-exhaustive technique, but it is fast, repeatable, and easy to perform. The training of operators and the learning curve is short (albeit rigorous) and can be used from physicians of different background and specialties. FoCUS should not be considered as a replacement of echocardiographic examination but as a clinical tool similar to a stethoscope to aid early diagnosis (at the bedside), define the event pathophysiology and prognostic stratification, and direct to the appropriate therapy. In this setting and in specific clinical situations, it is extremely useful.

## Figures and Tables

**Figure 1 medicina-55-00423-f001:**
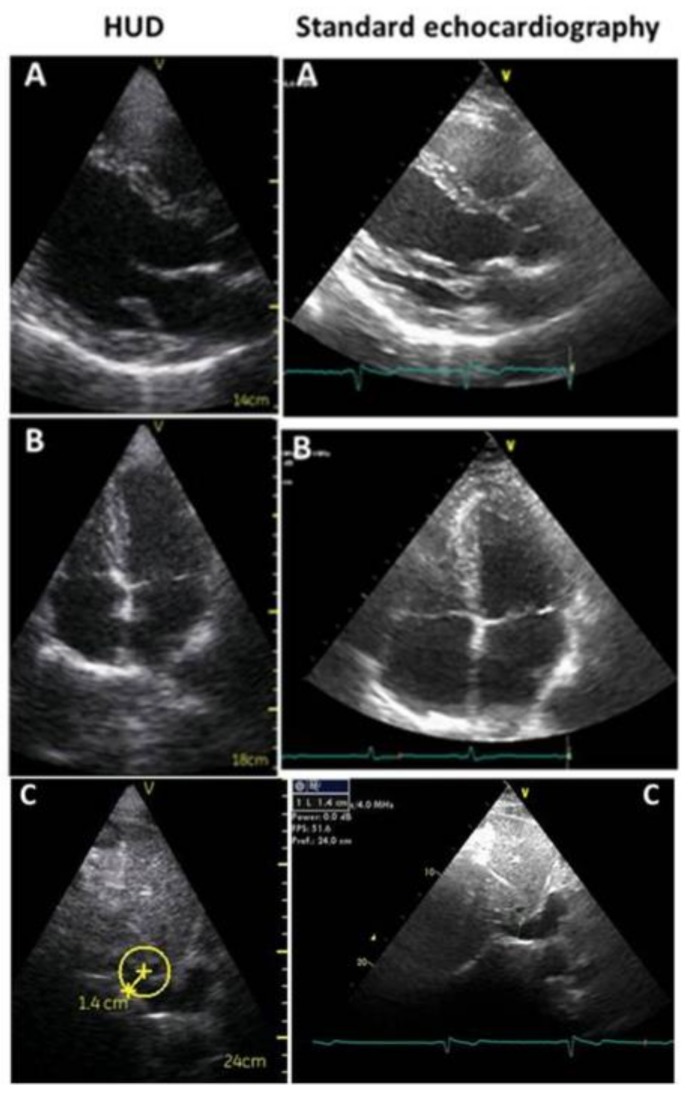
Echocardiographic images acquired in the same patient, both with HUD (on the **left**) and standard echocardiography (on the **right**). (**A**): parasternal long axis view, (**B**): apical four chambers view, (**C**): subcostal view for inferior vena cava. HUD defines qualitatively normal ventricular diameters and wall thicknesses, and mild left and right atrium dilatation. HUD measurement is accurate (see **C**). In standard echocardiography endocardium, mitral and aortic valves are better visualized.

**Figure 2 medicina-55-00423-f002:**
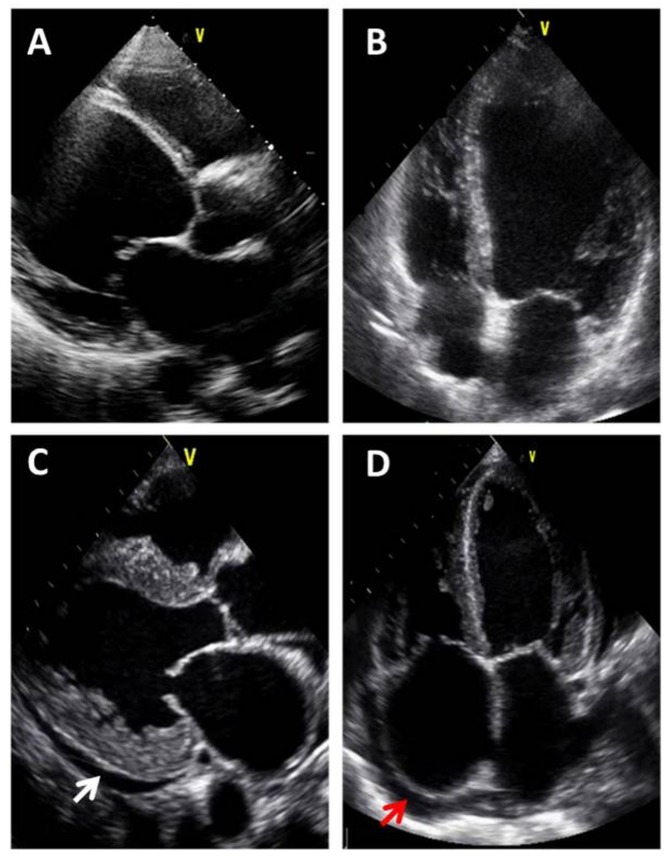
FoCUS examples in clinical suspicion of congestive heart failure. Parasternal long axis view (**A**) and apical four chambers view (**B**) in a patient with dilated cardiomyopathy and systolic heart failure: left ventricle is markedly dilated, wall thicknesses are normal, and ejection fraction is reduced. Parasternal long axis view (**C**) and apical four chambers view (**D**) in a patient with hypertrophic cardiomyopathy and diastolic heart failure: left ventricle is normal, wall thicknesses have increased, both left and right atrium are dilated, and mild pericardial effusion is present (arrows).

**Figure 3 medicina-55-00423-f003:**
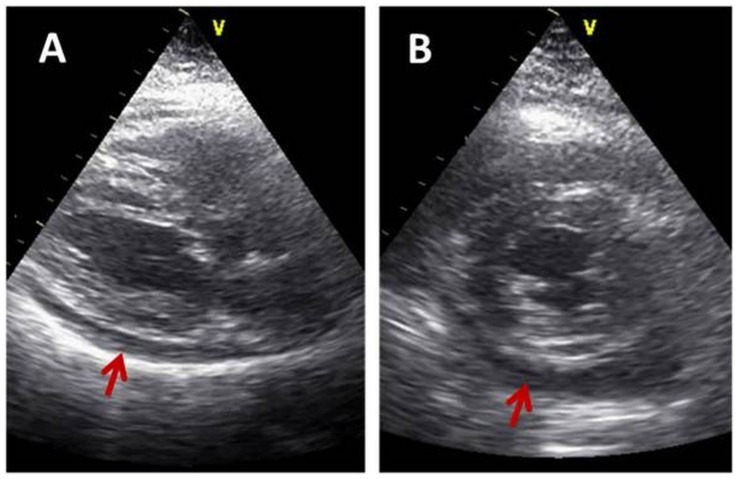
Parasternal long axis view (**A**) and left ventricle short axis view (**B**) in a patient with pericarditis. Arrows show a moderate pericardial effusion.

**Table 1 medicina-55-00423-t001:** Differences between standard and FoCUS echocardiography.

	Standard Echocardiography 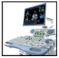	Focused Echo 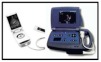
**Equipment**	Stationary, top level technology	Handheld ultrasound devices
**Echo purpose**	All heart diseases study	Few clinical questions
**Echo goal**	Accurate study	Differential diagnosis
**US sections**	All recommended	Few
**Execution time**	>30 min	Few minutes
**Diagnosis**	In election	Rapid
**Operator training**	Accurate/long	Rapid/25–50 exams

**Table 2 medicina-55-00423-t002:** FoCUS diagnostic capabilities in stable patients.

Clinical Suspect	FoCUS Goals
Congestive heart failure	Left atrium and ventricle dimension, LV EF, significant valvular regurgitation, IVC dimension and collapse, pulmonary congestion
Cardiomyopathies	Left ventricular hypertrophy, dilated cardiomyopathies
Atrial fibrillation	Left atrial dimensions, left ventricular function
Stable coronary artery disease	Wall motion abnormalities, ejection fraction
Valvular stenosis	Thickness and calcification leaflet, reduced leaflets mobility, turbulent transvalvular flow at color-Doppler
Valvular regurgitation	Extent regurgitation jet at color-Doppler
Pericarditis	Pericardial effusion presence, size and distribution

LV: left ventricle, EF: ejection fraction, IVC: inferior vena cava.

**Table 3 medicina-55-00423-t003:** POCUS diagnostic capabilities in unstable patients.

Clinical Scenarios	POCUS Goal	Rule-In/Rule-Out Diseases
Acute respiratory failure	No lung sliding sign (lung movement absence), Anechoic space between visceral and parietal pleura, Lines B—comets	Pneumotorax Pleural effusion Pulmonary congestion
Shock	Chambers size and shape, LV EF, IVC size and collapsibility, pericardial effusion, pulmonary congestion	Cardiogenic shock
Cardiac tamponade	Pericardial fluid size and distribution Guide to pericardiocentesis	Massive pericardial effusion
Acute chest pain	LV wall motion abnormalities, LV EF, RV size and function, Ascending aorta dimensions and morphology, aortic regurgitation, pericardial effusion	Acute coronary syndrome Pulmonary embolism Aortic dissection Pericarditis
Arrhythmic cardiac arrest	Left ventricular hypertrophy, LV dilatation, LV dysfunction, RV dilatation and dysfunction	Hypertrophic cardiomyopathy Dilated cardiomyopathy Arrhythmogenic right ventricular cardiomyopathy
Cardiopulmonary resuscitation	Massive pericardial effusion, RV dilatation, RV severe dysfunction, LV severe dysfunction, LV very small dimension (empty ventricle)	Cardiac tamponade Massive pulmonary embolism Severe congestive heart failure Hypovolemia

LV: left ventricle, EF: ejection fraction, IVC: inferior vena cava, RV: right ventricle.

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
