# Peer review of "Handheld Ultrasound and Focused Cardiovascular Echography: Use and Information"

_medicina, 2019, doi:10.3390/medicina55080423_

Round 1
Reviewer 1 Report
The manuscript presents the use of HUD (handheld ultrasound devices) devices and describes a basic comparison with others equipment. Several clinical scenarios are briefly presented, in stable and unstable patients, illustrating the interest in using HUD.
I believe the investigation is worthwhile and provides insightful information for the community, but the raw nature, inconsistencies, missing information, deeper comparison of the methods, lack of a detailed clinical case illustrating the advantages/limitations of each equipment use, makes the scientific contribution of the manuscript impossible to assess.
Starting from the title, the author must include the clinical application: cardiovascular applications. We understand this from the manuscript, but this must be clearly stated in the title. This is not a general review, for all kind of applications.
The biography is not complete. Other publications, such as https://doi.org/10.1016/j.ijcard.2018.12.014, https://www.ncbi.nlm.nih.gov/pubmed/21073515 and others deserved to be mentioned here because they are treating the same subject. In addition, in introduction section, references for different modalities (2D, M-mode, Doppler, etc) are also missing. Refs for cardiomyopathies or cardiac tamponade, valvular regurgitation (section 3) are also missing.
My major concern is the lack of a practical example of one clinical case for cardiovascular application, where are illustrating the HUD acquisition versus a standard acquisition, of course using the same US parameters, if possible. A deeper comparison of the HUD versus other equipment must be included, the acquisition parameters must be discussed and also the comparison regarding the US applicators (number of elements of the transducer, frequency, number of frames per seconds, image quality, accuracy, etc). And a general, more complex, discussion must follow, taking into account all this ‘technical parameters’. Moreover, aspects like 3D acquisitions, motion tracking, must also be included for a complete and general comparison study.
Moreover, I consider the physician’s experience is a delicate point that deserves to be better detailed. The image quality is not excellent when using HUD; therefore a more experienced investigator is required, for a correct diagnostic.
A better explanation of the differences between POCUS and FoCUS must be added; in this current form it is hard to understand exactly the differences.
Other than that, I have other minor issues:
Due to the English quality and typing errors, I recommend a careful rereading of the manuscript. I am listing here only few errors:
- Page 1, line 35: “can be perform”
- Page 1, line 35: please delete “all” – not all the basic information for diagnostic are obtained…
- Page 1, line 38: “it use”
- Page 1, line 43 and after: “utilizing” please use “using”
- Page 1, line 43 “systems” used 2 times
- Page 2, line 54: HUD – already defined
- Page 2, line 56: “measurements available” – please inverse
- Page 3, line 81, line 89: IVC already defined
- Page 3, table: please define EF – ejection fraction (here is used for the first time)
- Page 3, line 103: “It provides define life-saving information…” – please reformulate
- Ref no 7 – please give the correct form of the reference.
In conclusion, at this point, I can’t recommend the publication of this manuscript in its current form.
Reviewer 2 Report
- The manuscript needs to be extensively revised by a professional English speaker. Not only grammatical errors, the language used does not sound scientific at some parts.
- The topic of the review is interesting and worth publishing. It is a comprehensive review that will be helpful for young physicians and one’s in training. It would have been more interesting if you summarized what has been recently published regarding the sensitivity and specificity of HUS and comparing to with another US equipment, if any.
- Please talk more regarding the physics of the HUS compared to conventional US and its limitations.
- It would be interesting if you add images for the same pathology as it appeared on HUS and conventional US, if possible.
- Please add more references, at least total of 30.
Author Response
The manuscript needs to be extensively revised by a professional English speaker: Manuscript rereading with thanks.
It would have been more interesting if you summarized what has been recently published regarding the sensitivity and specificity of HUS and comparing to with another US equipment, if any: Comment of HUD versus standard echo are added (References 7, 8,18)
Please talk more regarding the physics of the HUS compared to conventional US and its limitations: Comment of HUD versus standard echo are added (Page 2 line 65)
It would be interesting if you add images for the same pathology as it appeared on HUS and conventional US, if possible: Images added with thanks (Page 2 line 76 HUD vs standard echocardiography - Page 4 line 123 dilated vs hypertrophic cardiomyopathy –
Page 5 line 140 pericardial effusion)
Please add more references, at least total of 30: References added with thanks. Total references: 37

Round 2
Reviewer 1 Report
Dear Authors,
The presented manuscript was improved compared to the first submission and the authors responded to my concerns.
But I still have few small remarks:
Figure 1 – please add a scale bar for the standard echocardiography images, for a better comparison.
Figure 2 – please add a scale bar and mention also the acquisition mode (HUD, FoCUS)
Figure 3 – please add a scale bar.
Page 3 – line 96 – please explain “”TTE”
Figure 3 – not explained in the text.
Reviewer 2 Report
authors respond to all my comments
Author Response
I thank the reviewer 2